materials science/nanotechnology

nanogel, pAA, BIS, high swelling degree, pH sensitivity

**Author for correspondence:**
Marcin Karbarz
e-mail: karbarz@chem.uw.edu.pl

This article has been edited by the Royal Society of Chemistry, including the commissioning, peer review process and editorial aspects up to the point of acceptance.

# Synthesis of cross-linked poly(acrylic acid) nanogels in an aqueous environment using precipitation polymerization: unusually high volume change

## Marcin Mackiewicz, Zbigniew Stojek and Marcin Karbarz

Faculty of Chemistry, Biological and Chemical Research Center, University of Warsaw, 101 Żwirki i Wigury Avenue, 02-089 Warsaw, Poland

MK, 0000-0002-7813-0513

For the first time, by using precipitation polymerization in an aqueous solution, a cross-linked poly(acrylic acid)—(pAA) nanogel was synthesized. pAA was synthesized and cross-linked with $N,N'$-methylenebisacrylamide (BIS) at 70°C in an acidified environment (pH 2) and containing 0.7 M NaCl using potassium persulfate as the initiator. Ionized pAA was soluble in water. The use of sodium chloride at low pH caused a decrease in the solubility of pAA and led to its precipitation and formation of cross-linked pAA nanogel. By using electron microscopies and light scattering techniques, the morphology, pH sensitivity and zeta potential of the obtained p(AA-BIS) nanogel were evaluated. The polymerization in an aqueous environment resulted in a very big swelling/shrinking coefficient (of approx. 4000) in response to pH and exhibited an unusually high negative zeta potential (of approx. −130 mV). These properties make the nanogel a very interesting sorbent and a construction material.

## 1. Introduction

Micro- and nanogels are cross-linked polymers of micro–nanosize filled with a solvent. They can exist either in the swollen or the shrunken state depending on the environmental conditions [1–4]. They exhibit very high surface areas, low viscosities and, compared with regular gels, can be injected into the body. They are characterized by a rapid and large-magnitude volume phase transition in response to changes in environmental conditions

such as temperature, ionic strength, pH and the redox state of the polymer [5–9]. Gels' sensitivity to environmental parameters is dependent on their functional components. Micro- and nanogels have great potential in applications in areas such as drug carriers, [10–13] catalysis [14,15] and sensors [16,17]. Micro- and nanogels containing carboxylic groups are capable of further modification and loading of active substances.

Cross-linked poly(acrylic acid)—p(AA) particles are examples of nanogels sensitive to pH. Due to the presence of a high amount of carboxylic groups, pAA gels undergo a big-volume transition in response to appropriate pH changes [18]. At pH smaller than $pK_a$ of pAA (4.5) [19], they exist in the shrunken state (carboxylic groups are protonated), while at pH higher than $pK_a$ they are swollen (carboxylic groups are dissociated).

The soft, swollen network of pAA nanogels, their hydrophilic nature and viscous consistency are similar to biological tissues. Additionally, their surface tension in biological fluids is low, they are biocompatible and have a reticulated structure. Because of these properties pAA gels are used in medicine as anticorrosion films for covering implants, as skin disease- or care materials, as high-bonding-strength medical glue [20–22] and especially as drug carriers [23]. The ability of pAA to adhere, for a long time, to biological tissues (bio(muco)adhesion) increases the drug absorption in the target site [24]. pAA gels are used in bioadhesive pharmaceutical hydrogels and sold as Polycarbophil or Carbopol [25,26]. The important property of pAA gels is that their polyanionic charge allows for easy and highly efficient loading of cationic substances [27] and allows them to combine with proteins and enzymes.

pAA nanogels are synthesized mainly by using microemulsion polymerization and distillation polymerization in a two-phase organic liquid/water medium with the help of stabilizers [28,29]. Ahmed *et al.* synthesized pAA nanogels via inverse-phase microemulsion polymerization at 70°C in liquid paraffin [30]. Tween 80 and Span 80 served as nonionic surfactants and azobisisobutyronitrile (AIBN) served as the initiator. The aqueous phase contained AA and *N,N′*-methylenebis(acrylamide) (BIS) as the cross-linking agent. These nanogels were used to adsorb cellulase, to enhance cellulase's efficacy and stability. In other work, Seyedeh *et al.* synthesized pAA nanogel to be used as a drug carrier. They used the distillation precipitation polymerization process with BIS as the linker [31]. Acetonitrile and AIBN were used as the solvent and the initiator, respectively. The reaction was performed at the boiling point of acetonitrile. We want to emphasize that, up to now, there is no report on the successful synthesis of pAA nanogel in just water.

As an environmentally-protective measure, the synthesis should be carried out in water and this was only partially possible. Acrylic acid was added as the comonomer during the emulsion polymerization of *N*-isopropylacrylamide (NIPA) and *N,N*-diethylacrylamide (DEA) [32–37]. Unfortunately, the molar content of AA in these gels did not exceed 40%. Moreover, the majority of micro- and nanogels containing ionized comonomers did not exhibit a high swelling degree. Even in the case of pure pAA, despite the fact that they consisted mainly of ionized groups, the swelling degree was not so high and their zeta potential did not exceed −60 mV [38,39].

The aim of this paper was to synthesize pure pAA nanogels in aqueous solution without the use of any toxic surfactants and stabilizers. We managed to synthesize pAA nanogel cross-linked with BIS using the precipitation polymerization process carried out in an aqueous environment without the addition of surfactants. We used sodium chloride and maintained low pH during the polymerization process to decrease the solubility of pAA and to precipitate it to form the nanogel. The nanogel exhibited very high volume change in response to pH and a very negative charge (zeta potential was approx. −130 mV at pH higher than $pK_a$). The reported values in the literature are much less negative [40–47].

# 2. Experimental

## 2.1. Chemicals

Acrylic acid (AA, 99%), potassium persulfate (KPS, 99.99%) and *N,N′*-methylenebis(acrylamide) (BIS, 99%) were supplied by Aldrich. NaOH (99%), HCl (35–38%) and NaCl (99%) were purchased from POCh. All ingredients were used as received. High-purity water was used for the preparation of all solutions. Water was purified with a Hydrolab/HLP purification system; the final conductivity of the water was 0.055 µS cm$^{-1}$.

## 2.2. Nanogel synthesis

The precipitation polymerization process, used in the synthesis of other smart polymers [48], was selected by us to obtain poly(AA-BIS) nanogels in aqueous media (scheme 1). AA and BIS were dissolved in 50 ml

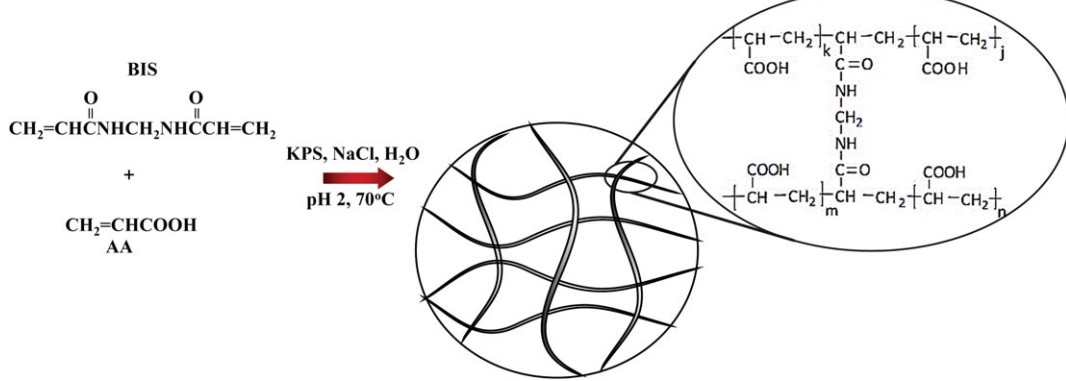

**Scheme 1.** Synthesis of poly(AA-BIS) nanogels.

of deionized water and placed in a flask equipped with a reflux condenser, an inlet- and an outlet of argon and a magnetic stirrer. Next, NaCl was added and the pH was set to a value of 2 by adding an appropriate amount of HCl to decrease the solubility of pAA. The final concentration of sodium chloride was 0.7 M. Other pH and ionic strength did not lead to better results. The mixture of reactants was heated at 70°C and deoxygenated for 30 min. Next, 13.5 mg of KPS was added to initiate polymerization. The molar linker/ AA ratio and the total concentration of the monomers were optimized to get acceptable polydispersity with the monomodal size distribution of the particles. We found that the molar linker/AA ratio of 0.01 and the total concentration of the monomers of 420 mM were optimal.

The reaction was carried out for 24 h under a blanket of argon. The stirring rate was set at 250 r.p.m. The unstable, bigger particles formed a sediment (it took 1 day) and in this way were removed. The stable fraction of the nanogel was purified by using glass wool and finally by dialysis. In the dialysis process, dialysis bags of 10 kDa molecular weight cut-off (Spectra/Por®) were used. The gel samples were dialysed with 5 l of water for three weeks at room temperature; water was changed daily. The yield of pAA nanogels, as compared to the initial AA monomer quantity, was approximately 50%. It was obtained gravimetrically after the separation of the purified nanogels.

## 2.3. Instrumental

### 2.3.1. Dynamic light scattering

A Malvern Zetasizer instrument (Nano ZS, UK) equipped with a 4-mW Helium-Neon laser with a wavelength of 632.8 nm and a scattering angle of 173° was used to determine the hydrodynamic diameter of the nanogel particles. The samples were kept at selected temperatures for 5 min before the measurements.

The pH of aqueous suspensions of nanogels was adjusted by adding either NaOH or HCl and was monitored just before the measurements using a Mettler Toledo, model SevenGo-SG2 pH-meter. Ionic strength was kept constant (0.09 M).

### 2.3.2. Electron microscopy (SEM and TEM)

A Merlin (Zeiss) instrument working at 3 kV was used to take SEM micrographs. A drop of the nanogel emulsion of pH *ca* 3 was placed on the SEM-sample holder and left to dry at RT. Before the measurements, the samples were coated with a 5-nm layer of the Au–Pd alloy deposited in vacuum using a Mini Sputter Coater (Polaron SC7620).

TEM micrograms were taken with a Libra 1200 (Zeiss) instrument. The accelerating voltage was 120 kV. A drop of nanogel dispersion of pH *ca* 3 was placed on a formvar-coated copper grid. Then samples were left to dry at RT. Brightfield (the contrast mode) was used for taking both SEM and TEM micrographs.

# 3. Results and discussion

The morphology of dried p(AA-BIS) nanogels was examined with electron microscopies. TEM and SEM micrographs of the dried nanogel emulsion of pH *ca* 3 are presented in figure 1*a,b*, respectively. At pH 3,

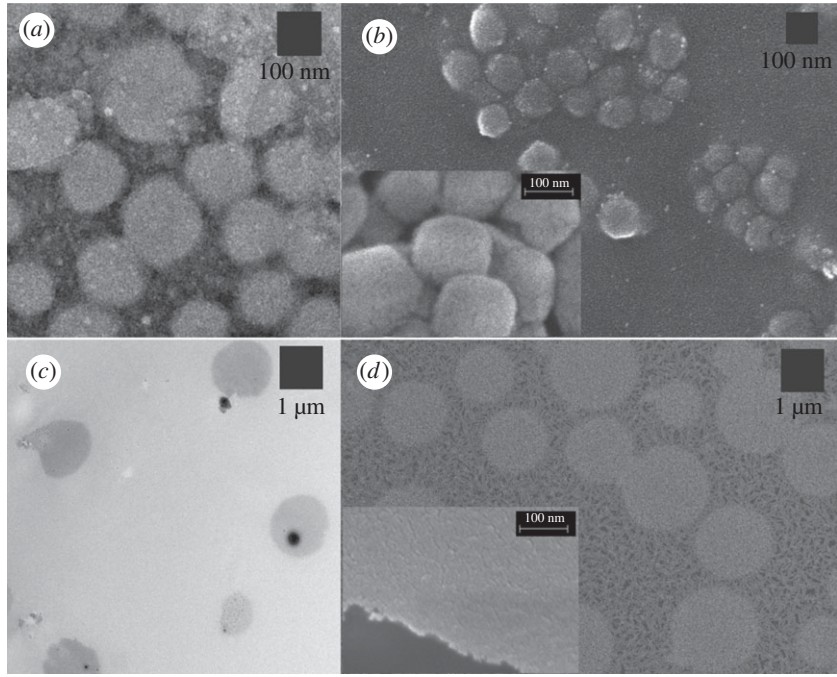

**Figure 1.** TEM (*a,c*) and SEM (*b,d*) micrographs of dried p(AA-BIS) nanogels at pH 3 (*a,b*) and pH 11 (*c,d*). Insets in (*b,d*): higher magnification allowing presentation and comparison of porosity of particle surfaces. SEM pictures were taken using mixed detectors: In-Lens and HE-SE2.

almost all carboxylic groups are protonated and nanogels are shrunken. As can be seen in the micrographs, the nanogel particles were more or less spherical. The mean sizes of the particles obtained from the SEM and TEM micrographs at pH 3 equalled $99 \pm 11$ (30 particles) and $137 \pm 22$ nm (13 particles), respectively. The micrographs of dried nanogel emulsion of pH *ca* 11 are presented in figure 1*c,d*. At pH 11 the nanogels were swollen (carboxylic groups were dissociated). The nanogel particles were spherical and their size was greater than 1 µm. The mean sizes of the particles obtained from the SEM and TEM micrographs equalled $1523 \pm 260$ nm (15 particles) and $1293 \pm 293$ nm (5 particles), respectively.

Next, the sensitivity of the nanogel to pH was examined using the DLS technique. The plot of the hydrodynamic diameter of the p(NIPA-BIS) nanogel as a function of pH is presented in figure 2*a*. Since $pK_a$ of carboxylic groups in pAA chains is approximately 4.5 [19], therefore at pH lower than $pK_a$, the diameter of the nanogel particles should be smaller than the diameter at pH above $pK_a$. As can be seen from figure 2*a*, the diameter of the nanogel did not change significantly below pH 4 and above pH 7. The biggest changes in the size of the nanogel were observed at pH in the range of 4–7. Apparently, the nanogel exhibited very high swelling changes. For example, the hydrodynamic diameter of the nanogel measured at pH 2 and pH 8 equalled 122 nm and 1990 nm, respectively. So, the change in the volume of the nanogel was more than three orders of magnitude; the nanogels swelled more than 4000 times. This was calculated by dividing the third powers of hydrodynamic diameters of the particles at pH 8 and 2 $(1990 \text{ nm})^3/(122 \text{ nm})^3$. Moreover, the nanogels were stable for a long time; therefore, for example, the sizes of particles measured at pH 2 and pH 8 after approximately one month were $131 \pm 60$ nm and $1897 \pm 190$ nm, respectively. Importantly, the hydrodynamic diameter is the size of particles with the hydration shells. Thus, this size of particles is rather bigger than the size of dried particles measured by using SEM and TEM. Higher magnification of the particles is presented in the insets in figure 1*b,d* to allow comparison of the porosity of the nanoparticles in pH 2 and 8.

Nanogel in the shrunken state, at acidic pH, formed a turbid aqueous emulsion, while its solution in the swollen state, at alkaline pH, was transparent. pH increase from 2 to 8 caused the polydispersity index to increase from 0.10 to 0.25. Typical size distributions of the nanogels measured at two selected pHs are presented in figure 2*b*.

The high swelling degree of the nanogel was related to the high amount of carboxylic groups in the polymer network and the corresponding, after dissociation, high negative charge. We investigated the

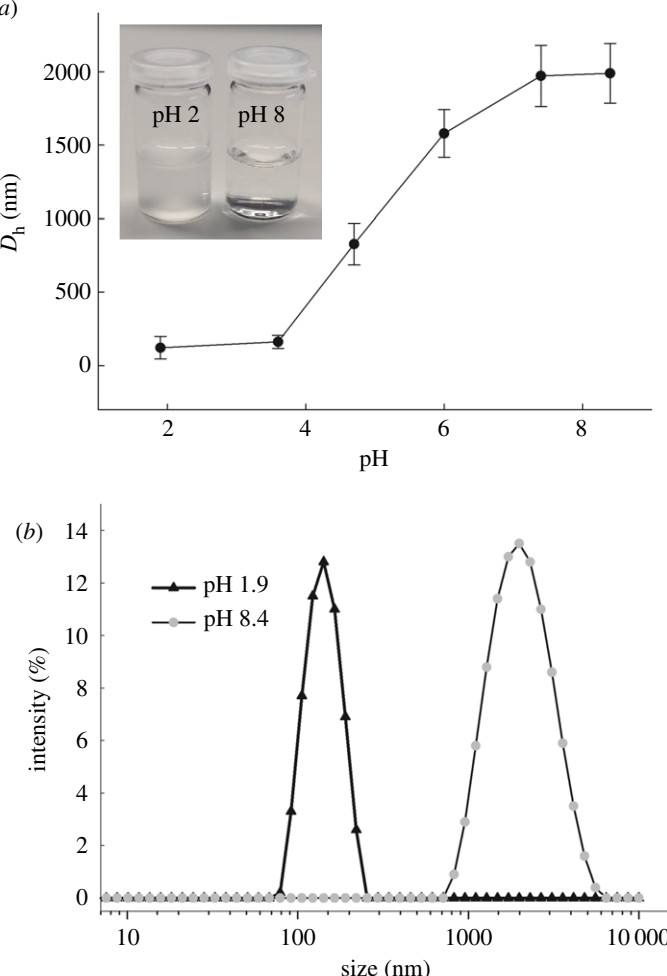

**Figure 2.** (*a*) Hydrodynamic diameter ($D_h$) of p(AA-BIS) nanogel (at 25°C) plotted versus pH. (*b*) Micro-/nanogel size distributions measured at two selected values of pH. Inset—optical images of nanogel emulsions in the shrunken and swollen states taken at two selected pHs. Ionic strength was kept constant at 90 mM.

zeta potential of the nanogel at different pHs. The obtained data are shown in figure 3. As can be seen, the nanogel exhibited an unusually big change in the zeta potential as the pH was increased. In an acidic environment, most of the carboxylic groups were protonated and the zeta potential was close to 0. A drastic decrease in the zeta potential (approx. −130 mV) was observed in the range of pH from approximately 4–8, where the carboxylic groups were deprotonated. The zeta potential dropped so much with that increase in pH because of the ionization of carboxylic groups and thus the corresponding increase in the negative charge.

## 4. Conclusion

We have demonstrated that with the application of proper conditions, there is a possibility for the synthesis of the cross-linked pAA nanogel in the aqueous environment. The obtained pH-sensitive p(AA-BIS) nanogel consisted of poly(acrylic acid) cross-linked with *N,N'*-methylenebisacrylamide. The precipitation polymerization process was used. It was carried out at 70°C and at pH 2 in water containing sodium chloride. pH 2 and the presence of sodium chloride caused a substantial decrease in the solubility of pAA, and thus enabled us to obtain the insoluble, cross-linked pAA nanogel. Thanks to conducting the polymerization in the aqueous solution, the obtained spherical particles of the nanogel exhibited unusually high sensitivity to pH. In the swollen state, the particle diameter was approximately 14 times bigger (changed from 122 nm at pH 2 to 1990 nm at pH 8) than in the shrunken state. So, the nanogel swelled more than 4000 times when the environment was changed from acidic to basic. This number can be compared with the swelling/shrinking ratio of the nanogels obtained in the traditional organic medium,

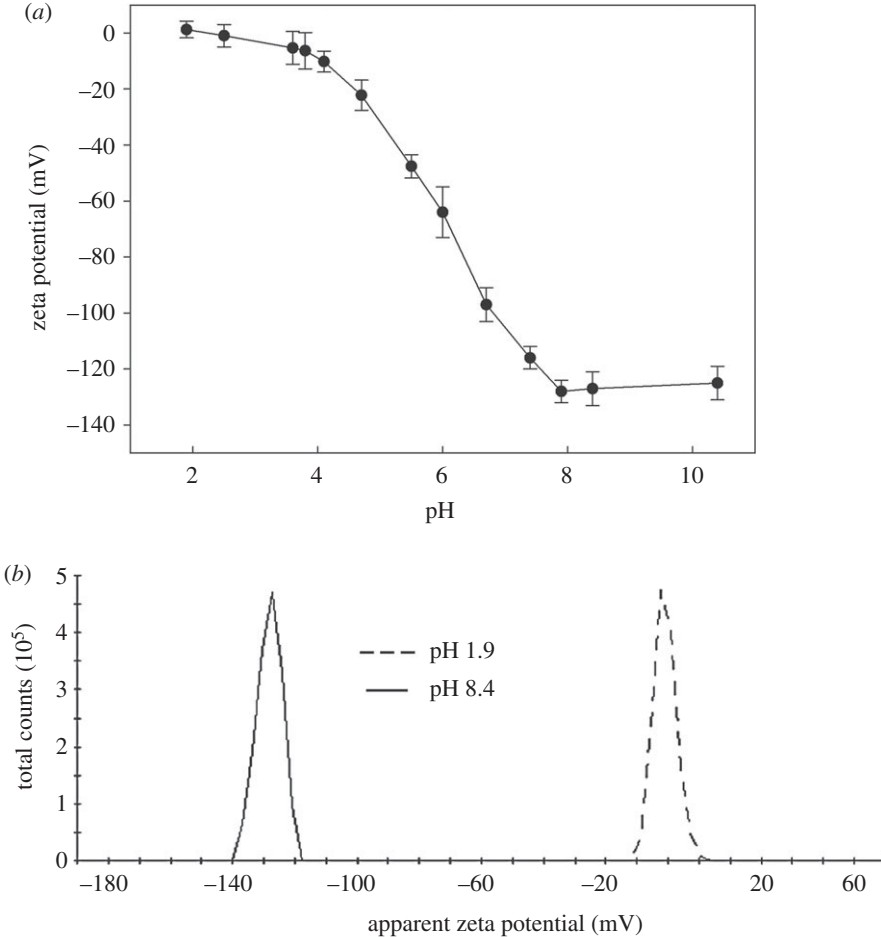

**Figure 3.** (*a*) Zeta potential of p(AA-BIS) nanogel measured versus pH at 25℃. Ionic strength was kept constant at 90 mM. Standard deviations indicated in the plot were estimated from three independent measurements. (*b*) Distribution of zeta potential measured for nanogel at two selected pH values.

where it was limited to the range of 7–25 [30,31,49,50]. The reason for this difference could be the presence of organic impurities in the gels. We want to add that the nanogel obtained in the aqueous environment also exhibited an unusually negative zeta potential at basic pH; it reached approximately −130 mV. In the future, we want to use this nanogel as a high-loading-capacity drug nanocarrier sensitive to pH.

Data accessibility. Raw data used to draw figures 2 and 3 have been uploaded as the electronic supplementary material.
Authors' contributions. M.M. carried out the laboratory work, collected field data, participated in data analysis, participated in the design of the study and drafted the manuscript; M.K. designed the study, coordinated the study, participated in data analysis, critically revised the manuscript and invented the main idea of paper and helped draft the manuscript; Z.S. critically revised the manuscript. All authors gave final approval for publication and agree to be held accountable for the work performed therein.
Competing interests. The authors declare no competing interest.
Funding. This work was supported by the National Science Center of Poland (grant no. 2015/19/B/ST5/03530). M.M. acknowledges financial support from the Ministry of Science and Higher Education in Poland.

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
