## [Reviewer comments · Royal Society Open Science]

Review History

RSOS-190981.R0 (Original submission)

Review form: Reviewer 1

Is the manuscript scientifically sound in its present form?

Yes

Are the interpretations and conclusions justified by the results?

Yes

Is the language acceptable?

Yes

Do you have any ethical concerns with this paper?

No

Have you any concerns about statistical analyses in this paper?

Yes

Recommendation?

Accept with minor revision (please list in comments)

Comments to the Author(s)

- The authors did not provide a total number of measurements for the TEM and SEM particle sizes and no standard deviation. Please change that.
- On Page 4 line 34 a referenc elink got broken. Please fix!
- Experimental part: Please provide the accelerating voltage of the TEM you used
- Experimental part: Please provide the contrast mode you used for taking the SEM micrographs. Especially figure 1 b and d look to my eye to be acquired by different contrast modes. It would be also appropriate to replace 1d with a better SEM micrograph which gives more information on the surface morphology. To the referee's eye 1d more looks like a TEM micrograph, but I could be wrong.
- Experimental part: Please provide the contrast mode you used for taking the TEM (bright field, dark field etc) micrographs. Especially figure 1 a and c look to my eye to be acquired by different contrast modes.
- Are the swollen particles porous? Please provide BET surface area of swollen and non-swollen samples.
- please make formatting consistens in 3A and 3B
- In some parts of the manuscript the hydrodynamic diameter of the particles is 122nm in others it is 142 nm. Please make that consisten.

Review form: Reviewer 2

Is the manuscript scientifically sound in its present form?

Yes

Are the interpretations and conclusions justified by the results?

Yes

Is the language acceptable?

Yes

Do you have any ethical concerns with this paper?

No

Have you any concerns about statistical analyses in this paper?

No

Recommendation?

Accept with minor revision (please list in comments)

Comments to the Author(s)

The authors describe a new strategy to fabricate PAA nanogels in an aqueous environment. The volume and zeta potential change of the particles is very intriguing. I recommend minor revisions as follows:

1. The authors should explain why the zeta potential drops so much with increase in pH.
2. Authors comment that the very negative charge is "never seen before". such a statement should be backed by a citation that reports the lowest charge seen.

3. The authors should perform size analysis from the SEM and TEM images and compare with DLS to see if during the drying at RT during EM sample prep there is no change from the true sample size.
4. Why did the authors not perform cryo-TEM of the nanogel?
5. It is a very short manuscript for a full paper. It is more of a communication. the authors should show data about the stability of the nanogels as a function of pH and time.
6. The authors should contemplate on possible applications of these nanogels given the extreme negative charge and drastic volume change.

Review form: Reviewer 3

Is the manuscript scientifically sound in its present form?

No

Are the interpretations and conclusions justified by the results?

No

Is the language acceptable?

No

Do you have any ethical concerns with this paper?

No

Have you any concerns about statistical analyses in this paper?

No

Recommendation?

Reject

Comments to the Author(s)

1. English should be improved
2. Page 6, line 49: "The optimization procedure revealed that the best molar linker/AA ratio was 0.01 and the best". It is not clearly explained how the optimization procedure was performed. What was optimized through this procedure?
3. It is not clear how the mean size of the particles shown in Figure 1 was calculated. How many samples were measured and what is standard deviation? Moreover, the paper claims that the nanogel particles were "of size from 90 to 130 nm" while on SEM pictures (Figure 1A) particles with diameter larger than 200 nm are clearly seen. Here, a size distribution of the nanoparticles should be presented obtained via SEM or TEM pictures, based on the diameters of at least 100 particles. Then the mean size should be claimed to be real. Moreover, Figure 1B is claimed to be TEM but it definitely is from SEM.
4. Page 10, line 8: "pKa of pAA chains" does not have any sense. pKa is for the COOH group
5. Page 10, line 20: It is not clear how these 4000 times of swelling were calculated: "change in the volume of the nanogel was more than three order of magnitude (the nanogels swell more than 4 000 times)"
6. The authors do not discuss what is the yield of PAA nanogels as compared to the initial AA monomer quantity?
7. In the Conclusions section the sizes provided for the same nanogels are different from those cited in the results and discussion section (difference being more than 20 nm).

8. Page 7, line 47: "The nanogel hydrodynamic diameter was measured in function of temperature" is a sentence from the DLS experiment description, but in the presented results no data about temperature dependence of the nanogels size are provided.

The paper has to be seriously worked on in order to be published.

Decision letter (RSOS-190981.R0)

22-Jul-2019

Dear Dr Karbarz:

Title: Synthesis of cross-linked poly(acrylic acid) nanogels in aqueous environment using precipitation polymerization. Unusually high volume change
Manuscript ID: RSOS-190981

The editor assigned to your manuscript has now received comments from reviewers. We would like you to revise your paper in accordance with the referee and Subject Editor suggestions which can be found below (not including confidential reports to the Editor). Please note this decision does not guarantee eventual acceptance.

Please submit your revised paper before 14-Aug-2019. Please note that the revision deadline will expire at 00.00am on this date. If we do not hear from you within this time then it will be assumed that the paper has been withdrawn. In exceptional circumstances, extensions may be possible if agreed with the Editorial Office in advance. We do not allow multiple rounds of revision so we urge you to make every effort to fully address all of the comments at this stage. If deemed necessary by the Editors, your manuscript will be sent back to one or more of the original reviewers for assessment. If the original reviewers are not available we may invite new reviewers.

Please also include the following statements alongside the other end statements. As we cannot publish your manuscript without these end statements included, if you feel that a given heading is not relevant to your paper, please nevertheless include the heading and explicitly state that it is not relevant to your work.

- Funding statement

Please include a funding section after your main text which lists the source of funding for each author.

RSC Associate Editor:
Comments to the Author:
(There are no comments.)

RSC Subject Editor:
Comments to the Author:
(There are no comments.)

Reviewers' Comments to Author:
Reviewer: 1

Comments to the Author(s)

- The authors did not provide a total number of measurements for the TEM and SEM particle sizes and no standard deviation. Please change that.
- On Page 4 line 34 a referenc elink got broken. Please fix!
- Experimental part: Please provide the accelerating voltage of the TEM you used
- Experimental part: Please provide the contrast mode you used for taking the SEM micrographs. Especially figure 1 b and d look to my eye to be acquired by different contrast modes. It would be also appropriate to replace 1d with a better SEM micrograph which gives more information on the surface morphology. To the referee's eye 1d more looks like a TEM micrograph, but I could be wrong.
- Experimental part: Please provide the contrast mode you used for taking the TEM (bright field, dark field etc) micrographs. Especially figure 1 a and c look to my eye to be acquired by different contrast modes.
- Are the swollen particles porous? Please provide BET surface area of swollen and non-swollen samples.
- please make formatting consistens in 3A and 3B

- In some parts of the manuscript the hydrodynamic diameter of the particles is 122nm in others it is 142 nm. Please make that consistent.

Reviewer: 2

Comments to the Author(s)

The authors describe a new strategy to fabricate PAA nanogels in an aqueous environment. The volume and zeta potential change of the particles is very intriguing. I recommend minor revisions as follows:

1. The authors should explain why the zeta potential drops so much with increase in pH.
2. Authors comment that the very negative charge is "never seen before". such a statement should be backed by a citation that reports the lowest charge seen.
3. The authors should perform size analysis from the SEM and TEM images and compare with DLS to see if during the drying at RT during EM sample prep there is no change from the true sample size.
4. Why did the authors not perform cryo-TEM of the nanogel?
5. It is a very short manuscript for a full paper. It is more of a communication. the authors should show data about the stability of the nanogels as a function of pH and time.
6. The authors should contemplate on possible applications of these nanogels given the extreme negative charge and drastic volume change.

Reviewer: 3

Comments to the Author(s)

1. English should be improved
2. Page 6, line 49: "The optimization procedure revealed that the best molar linker/AA ratio was 0.01 and the best". It is not clearly explained how the optimization procedure was performed. What was optimized through this procedure?
3. It is not clear how the mean size of the particles shown in Figure 1 was calculated. How many samples were measured and what is standard deviation? Moreover, the paper claims that the nanogel particles were "of size from 90 to 130 nm" while on SEM pictures (Figure 1A) particles with diameter larger than 200 nm are clearly seen. Here, a size distribution of the nanoparticles should be presented obtained via SEM or TEM pictures, based on the diameters of at least 100 particles. Then the mean size should be claimed to be real. Moreover, Figure 1B is claimed to be TEM but it definitely is from SEM.
4. Page 10, line 8: "pKa of pAA chains" does not have any sense. pKa is for the COOH group
5. Page 10, line 20: It is not clear how these 4000 times of swelling were calculated: "change in the volume of the nanogel was more than three order of magnitude (the nanogels swell more than 4 000 times)
6. The authors do not discuss what is the yield of PAA nanogels as compared to the initial AA monomer quantity?
7. In the Conclusions section the sizes provided for the same nanogels are different from those cited in the results and discussion section (difference being more than 20 nm).
8. Page 7, line 47: "The nanogel hydrodynamic diameter was measured in function of temperature" is a sentence from the DLS experiment description, but in the presented results no data about temperature dependence of the nanogels size are provided.

The paper has to be seriously worked on in order to be published.

Author's Response to Decision Letter for (RSOS-190981.R0)

See Appendix A.

RSOS-190981.R1 (Revision)

Review form: Reviewer 1

Is the manuscript scientifically sound in its present form?

Yes

Are the interpretations and conclusions justified by the results?

Yes

Is the language acceptable?

Yes

Do you have any ethical concerns with this paper?

No

Have you any concerns about statistical analyses in this paper?

No

Recommendation?

Accept with minor revision (please list in comments)

Comments to the Author(s)

Please add the SEM detectors you used in the description of the SEM pictures in Fig. 1 as described in the answer to the comments.

Decision letter (RSOS-190981.R1)

02-Sep-2019

Dear Dr Karbarz:

Title: Synthesis of cross-linked poly(acrylic acid) nanogels in aqueous environment using precipitation polymerization. Unusually high volume change

Manuscript ID: RSOS-190981.R1

Thank you for submitting the above manuscript to Royal Society Open Science. On behalf of the Editors and the Royal Society of Chemistry, I am pleased to inform you that your manuscript will be accepted for publication in Royal Society Open Science subject to minor revision in accordance with the referee suggestions. Please find the reviewers' comments at the end of this email.

The reviewers and handling editors have recommended publication, but also suggest some minor revisions to your manuscript. Therefore, I invite you to respond to the comments and revise your manuscript.

Please also include the following statements alongside the other end statements. As we cannot publish your manuscript without these end statements included, if you feel that a given heading is not relevant to your paper, please nevertheless include the heading and explicitly state that it is not relevant to your work. We have included a screenshot example of the end statements for reference.

- Acknowledgements

- Funding statement

Please include a funding section after your main text which lists the source of funding for each author.

Because the schedule for publication is very tight, it is a condition of publication that you submit the revised version of your manuscript before 11-Sep-2019. Please note that the revision deadline will expire at 00.00am on this date. If you do not think you will be able to meet this date please let me know immediately.

Best wishes,
Dr Laura Smith
Publishing Editor, Journals

RSC Associate Editor:
Comments to the Author:
(There are no comments.)

RSC Subject Editor:
Comments to the Author:
(There are no comments.)

Reviewer comments to Author:
Reviewer: 1

Comments to the Author(s)
Please add the SEM detectors you used in the description of the SEM pictures in Fig. 1 as described in the answer to the comments.

Author's Response to Decision Letter for (RSOS-190981.R1)

See Appendix B.

Decision letter (RSOS-190981.R2)

04-Oct-2019

Dear Dr Karbarz:

Title: Synthesis of cross-linked poly(acrylic acid) nanogels in aqueous environment using precipitation polymerization. Unusually high volume change
Manuscript ID: RSOS-190981.R2

It is a pleasure to accept your manuscript in its current form for publication in Royal Society Open Science. The chemistry content of Royal Society Open Science is published in collaboration with the Royal Society of Chemistry.

RSC Associate Editor
Comments to the Author:
The manuscript can now be accepted.

Reviewer(s)' Comments to Author:

Appendix A

Replies to Reviewers' criticism

We thank the reviewers for their useful comments and suggestions. We took all the comments into consideration and revised the paper accordingly. We feel the paper has been substantially improved.

Reviewer 1

1. - The authors did not provide a total number of measurements for the TEM and SEM particle sizes and no standard deviation. Please change that.

Author's response:

Number of measurements and standard deviation of the size were added to the text.

2. - On Page 4 line 34 a reference link got broken. Please fix!

Author's response:

It was fixed.

3. - Experimental part: Please provide the accelerating voltage of the TEM you used

Author's response:

Accelerating voltage was added to the text.

4. - Experimental part: Please provide the contrast mode you used for taking the SEM micrographs. Especially figure 1 b and d look to my eye to be acquired by different contrast modes. It would be also appropriate to replace 1d with a better SEM micrograph which gives more information on the surface morphology. To the referee's eye 1d more looks like a TEM micrograph, but I could be wrong.

Author's response:

In all micrographs bright field contrast mode was used. This information was added to the text. The pictures were taken in crossover mode. Picture 1B was taken using mixed detectors: In-Lens and HE-SE2, this probably led to a difference in contrast. An inset was added to Fig 1d to illustrate the morphology of the particles. We checked it: Figure 1D looks like a TEM but in fact it is a SEM micrograph.

5. - Experimental part: Please provide the contrast mode you used for taking the TEM (bright field, dark field etc) micrographs. Especially figures 1 a and 1 c look to my eye to be acquired by different contrast modes.

Author's response:

In all micrographs bright field contrast mode was used. The differences in contrast may be caused by the difference in density of the examined materials.

6. - Are the swollen particles porous? Please provide BET surface area of swollen and non-swollen samples.

Author's response:

Unfortunately we could not use the BET approach; however, now the porosities of the swollen and shrunk particles are illustrated by presenting appropriately magnified SEM micrographs. They were added as insets to Figures 1B and 1D.

7. - please make formatting consistent in 3A and 3B

Author's response:

It was done

8. - In some parts of the manuscript the hydrodynamic diameter of the particles is 122nm in others it is 142 nm. Please make that consistent.

Author's response:

We thank the reviewer. The numbers were corrected in the text.

Reviewer 2

1. The authors should explain why the zeta potential drops so much with increase in pH.

Author's response:

An explanation was added to the text.

2. Authors comment that the very negative charge is "never seen before". Such a statement should be backed by a citation that reports the lowest charge seen.

Author's response:

Appropriate references and more text were added to the ms.

3. The authors should perform size analysis from the SEM and TEM images and compare with DLS to see if during the drying at RT during EM sample prep there is no change from the true sample size.

Author's response:

This is a very useful comment. Yes, the size of aqueous dispersion of particles measured by DLS was rather bigger than that of dried particles measured by EMs. Some text was added to the ms.

4. Why did the authors not perform cryo-TEM of the nanogel?

Author's response:

Unfortunately we had no access to cryo-TEM. In fact we did not have to do it because cryo-TEM is especially useful for core-shell particles to visualize the differences between the core and the shell; but we worked with uniformly filled spheres. This uniform filling was well seen in the experiments where the particles were stained with uranyl acetate.

5. It is a very short manuscript for a full paper. It is more of a communication. the authors should show data about the stability of the nanogels as a function of pH and time.

Author's response:

The size of the particles measured at two selected pH did not change significantly after one month and these results were added to the text

6. The authors should contemplate on possible applications of these nanogels given the extreme negative charge and drastic volume change.

Author's response:

We noticed in the conclusions section that these properties make the nanogel a very interesting sorbent and a construction material. In the future we plan to use this nanogel as a potential high-loading-capacity drug nanocarrier sensitive to pH.

Reviewer 3

1. English should be improved

Author's response:

We have made an effort to improve English.

2. Page 6, line 49: "The optimization procedure revealed that the best molar linker/AA ratio was 0.01 and the best". It is not clearly explained how the optimization procedure was performed. What was optimized through this procedure?

Author's response:

The molar linker/AA ratio and the total concentration of the monomers were optimized to get acceptable polydispersity with monomodal size distribution of the particles. We found that molar linker/AA ratio of 0.01 and the total concentration of the monomers of 420 mM were optimal. This finding was added to the text.

3. It is not clear how the mean size of the particles shown in Figure 1 was calculated. How many samples were measured and what is standard deviation? Moreover, the paper claims that the nanogel particles were "of size from 90 to 130 nm" while on SEM pictures (Figure 1A) particles with diameter larger than 200 nm are clearly seen. Here, a size distribution of the nanoparticles should be presented obtained via SEM or TEM pictures, based on the diameters of at least 100 particles. Then the mean size should be claimed to be real. Moreover, Figure 1B is claimed to be TEM but it definitely is from SEM.

Author's response:

We added mean size and standard deviation calculated by using SEM and TEM images, and the text was corrected. After a careful analysis we reached a conclusion that Figure 1B is really a SEM micrograph.

4. Page 10, line 8: "pKa of pAA chains" does not have any sense. pKa is for the COOH group

Author's response:

It was corrected.

5. Page 10, line 20: It is not clear how these 4000 times of swelling were calculated: "change in the volume of the nanogel was more than three order of magnitude (the nanogels swell more than 4 000 times)"

Author's response:

The way of calculations was added to the text.

6. The authors do not discuss what is the yield of PAA nanogels as compared to the initial AA monomer quantity?

Author's response:

The yield of pAA nanogels was added to the text.

7. In the Conclusions section the sizes provided for the same nanogels are different from those cited in the results and discussion section (difference being more than 20 nm).

Author's response:

It was corrected.

8. Page 7, line 47: "The nanogel hydrodynamic diameter was measured in function of temperature" is a sentence from the DLS experiment description, but in the presented results no data about temperature dependence of the nanogels size are provided.

Author's response:

Indeed it was a mistake and it was corrected.

Appendix B

Replies to Reviewers' criticism

Reviewer 1

1. Please add the SEM detectors you used in the description of the SEM pictures in Fig. 1 as described in the answer to the comments.

Author's response:

Thank you. The information about detectors was added to the Fig. 1 description.